# Accelerating amorphous polymer electrolyte screening by learning to reduce errors in molecular dynamics simulated properties

Tian Xie [1,2✉], Arthur France-Lanord [1,3], Yanming Wang [1,3], Jeffrey Lopez [3], Michael A. Stolberg[1,4], Megan Hill[4], Graham Michael Leverick [5], Rafael Gomez-Bombarelli [1], Jeremiah A. Johnson [4], Yang Shao-Horn [1,5] & Jeffrey C. Grossman [1,3✉]

Polymer electrolytes are promising candidates for the next generation lithium-ion battery technology. Large scale screening of polymer electrolytes is hindered by the significant cost of molecular dynamics (MD) simulation in amorphous systems: the amorphous structure of polymers requires multiple, repeated sampling to reduce noise and the slow relaxation requires long simulation time for convergence. Here, we accelerate the screening with a multi-task graph neural network that learns from a large amount of noisy, unconverged, short MD data and a small number of converged, long MD data. We achieve accurate predictions of 4 different converged properties and screen a space of 6247 polymers that is orders of magnitude larger than previous computational studies. Further, we extract several design principles for polymer electrolytes and provide an open dataset for the community. Our approach could be applicable to a broad class of material discovery problems that involve the simulation of complex, amorphous materials.

[1] Department of Materials Science and Engineering, Massachusetts Institute of Technology, Cambridge, MA 02139, USA. [2] Computer Science and Artificial Intelligence Laboratory, Massachusetts Institute of Technology, Cambridge, MA 02139, USA. [3] Research Laboratory of Electronics, Massachusetts Institute of Technology, Cambridge, MA 02139, USA. [4] Department of Chemistry, Massachusetts Institute of Technology, Cambridge, MA 02139, USA. [5] Department of Mechanical Engineering, Massachusetts Institute of Technology, Cambridge, MA 02139, USA. ✉email: txie@csail.mit.edu; jcg@mit.edu

Polymer electrolytes are promising candidates for next-generation lithium-ion battery technology due to their low cost, safety, and manufacturing compatibility. The major challenge with the current polymer electrolytes is their low ionic conductivity, which limits the usage in real-world applications[1–3]. This limitation has motivated tremendous research efforts to explore new classes of polymers via both experiments[4–7] and atomic-scale simulations[8–10]. However, the simulation of ionic conductivity is extremely expensive due to the amorphous nature of polymer electrolytes and the diversity of timescales involved in their dynamics, drastically limiting the ability to employ high-throughput computational screening approaches. Note that although some polymers have crystalline structures and past studies have performed large-scale screenings on crystalline polymers with density functional theory calculations[11,12], screening polymers with lower levels of crystallinity requires more expensive molecular dynamics (MD) simulations to sample the equilibrium structure and dynamics. For instance, recent studies[8–10] exploring amorphous polymer electrolytes with classical MD only simulated around ten polymers. In contrast, a study that applies machine learning methods to literature data is able to explore a larger chemical space[7], but it is limited by the diversity of polymers that have been studied in the past. The exploration beyond known chemical spaces would require a significant acceleration of the computational screening of polymer electrolytes.

There are two major reasons for the large computational cost for simulating the ionic conductivity of polymer electrolytes with MD. First, the amorphous structure of polymer electrolytes can only be sampled from a random distribution using, e.g., Monte Carlo algorithms, and yet this initial structure has a significant impact on the simulated ionic conductivity due to the lack of ergodicity in the MD simulation[10,13]. Multiple simulations starting from independent configurations are therefore required in order to properly sample the phase space and reduce statistical noise. Second, the slow relaxation of polymers requires a long MD simulation time to achieve convergence for ionic conductivity (on the orders of 10's to 100's of ns), so each MD simulation is also computationally expensive[8,10].

Machine learning (ML) techniques have been widely used to accelerate the screening of ordered materials[14,15], but most previous studies implicitly[16–19] assume that the properties used to train the ML models are generated through a deterministic, unbiased process. However, the MD simulation of complex materials like amorphous polymers is intrinsically stochastic, and obtaining data with low statistical uncertainties by running repetitive simulations is impractical at a large scale due to the large computational cost. An alternative approach is to reduce the accuracy requirements for individual MD simulations and learn to reduce the random and systematic errors with large quantities of less expensive, yet imperfect data. It has previously been demonstrated that ML models can learn from noisy data and recover the true labels for images[20] and graphs[21]. Past works have also shown that systematic differences between datasets can be learned by employing transfer learning techniques[22–25]. Inspired by these results, we hope to significantly reduce the computational cost for simulating the transport behavior of polymers by adopting a noisy, biased simulation scheme with short, unconverged MD simulations.

In this work, we aim to accelerate the high throughput computational screening of polymer electrolytes by learning from a large amount of biased, noisy data and a small number of unbiased data from molecular dynamics simulations. Despite the large random errors caused by the dependence on the initial structure, we only perform one MD simulation for each polymer, and learn a shared model across polymers to reduce the random error and recover true properties that one would obtain from repetitive simulations. To reduce the long MD simulation time, we perform large quantities of short, unconverged MD simulations and a small number of long, converged simulations. We then employ multitask learning to learn a correction from the short simulation properties to long simulation properties. We find that our model achieves a prediction error with respect to the true properties smaller than the random error from a single MD simulation, and it also corrects the systematic errors from unconverged simulations better than a linear correction. Combining the reduction of both random and systematic errors, we successfully screen space of 6247 polymers and discover the best polymer electrolytes from the space, which corresponds to a 22.8-fold acceleration compared with simulating each polymer directly with one long simulation. Finally, we extract several design principles for polymer electrolytes by analyzing the predicted properties in the chemical space.

## Results

**Polymer space and sources of errors**. The polymer space we aim to explore is defined in Fig. 1a, which considers both the synthesizability of polymers and their potential as electrolytes. In general, it is difficult to determine the synthesizability, especially the polymerizability, of an unknown polymer. Here, we focus on a well-established condensation polymerization route using carbonyl dichloride and comonomers containing any combination of two primary hydroxyl, amino, or thiol groups to form poly-carbonates, ureas, dithiocarbonates, urethanes, thiourethanes, and thiocarbonates. This scheme does not guarantee polymerizability, but provides a likely route for lab synthesis. The carbonyl structure ensures a minimum capability to solvate Li-ions as an electrolyte, and it also allows for the maximum diversity of polymer backbones. The monomers are sampled from a large pharmaceutical database[26] to ensure its structures are realistic. After obtaining the molecular structure of the polymer, we sample its 3D amorphous structure with a Monte Carlo algorithm, insert 1.5 mol lithium bis(trifluoromethanesulfonyl)imide (LiTFSI) salt per kilogram of polymer, perform a 5 ns MD equilibration, and finally run the MD simulation to compute its transport properties like conductivity.

There are mainly two types of errors in this workflow. In the scope of this work, we call random errors the ones that can be eliminated by running repetitive simulations on the same polymer, and systematic errors those that cannot be eliminated. The major source of random error is the sampling of the initial amorphous structure of the polymer. In Fig. 1b, we show the conductivities computed from six different random initializations for the same polymer, which has a large standard deviation of 0.094 $\log_{10}$(S/cm) in the log scale at 5 ns. This error comes from the lack of ergodicity of MD simulations for polymers—the large-scale amorphous structure of the polymers usually does not change significantly at the timescale that can be achieved with MD. The systematic errors mainly come from the long MD simulation time needed to obtain the converged conductivity. Figure 1c shows the value of conductivity as a function of the simulation time for five different polymers, which slowly converges as the simulation progresses. This slow convergence introduces a systematic error of ionic conductivity with any specified simulation time with respect to the converged conductivity. On average, there is a 0.435 $\log_{10}$(S/cm) difference in the log scale between a 5 ns and a 50 ns simulation for these five polymers. Here, we use the 50 ns simulation results as the converged values, although it is not fully converged for some polymers. Based on our comparison with respect to experimental values reported in literature[4,6,27–36] in Supplementary Fig. 1, the

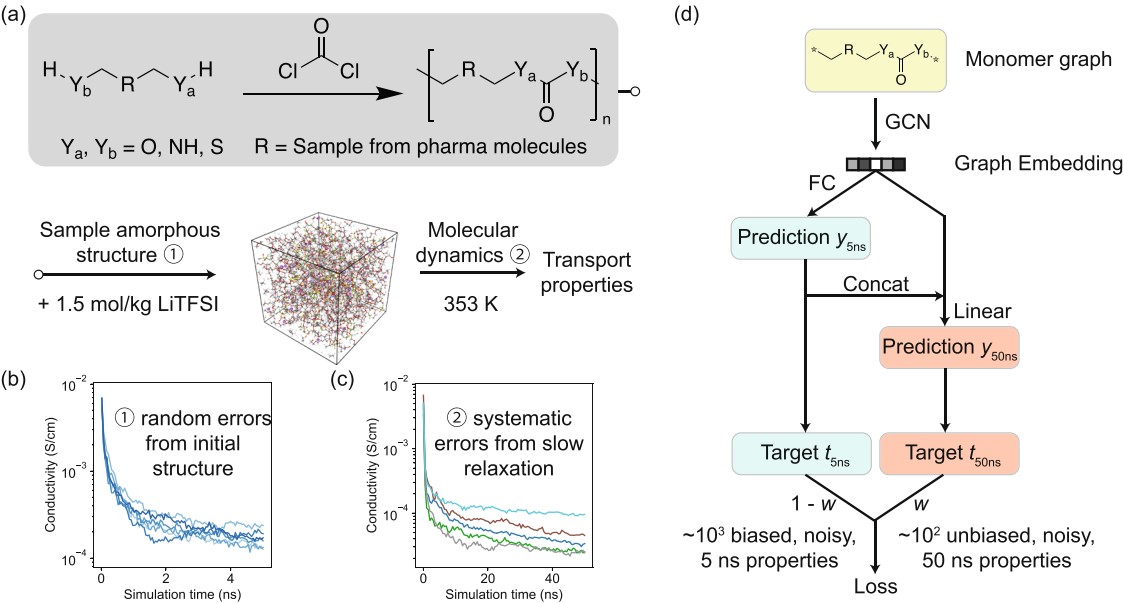

**Fig. 1 Illustration of the polymer space and the learning framework. a** The polymer space and molecular dynamics simulation workflow. **b** Ionic conductivity as a function as simulation time from six independent 5 ns MD runs for the same polymer, showing the random errors caused by the amorphous structure initialization. **c** Ionic conductivity as a function as simulation time for five different polymers, showing the long simulation time needed for convergence. (Polymer structures for (**b**, **c**) are provided in the supplementary information.) **d** Multitask learning framework to reduce the random and systematic errors from MD simulations.

50 ns simulation has a reasonable agreement except for polymers with very low conductivity. Note that even 50 ns conductivities have large random errors similar to the 5 ns conductivities, since the random errors are mainly caused by the large-scale amorphous structures that do not change significantly with long simulation time. In addition to the random and systematic errors, the difference between the 50 ns simulation and experimental results represents the simulation error of the MD approach, which is influenced by the accuracy of force field, finite size of the simulation box, etc. We do not consider this simulation error for most of our multitask learning workflow and only use experimental data for final evaluation. In principle, if we have enough experimental data, they can also be incorporated into the learning framework similar to the systematic error to further improve the prediction accuracy with respect to experimental results.

**Multitask model to reduce errors**. These two types of errors introduce significant computational costs to achieve an accurate calculation of ionic conductivity, because such a calculation requires repetitive simulations on the same polymer that are also individually expensive. Here, we attempt to reduce these errors by learning a shared model across the polymer space. To achieve this goal, we develop a multitask graph neural network architecture (Fig. 1d) to learn to reduce both random and systematic errors from MD simulations. We first encode the monomer structure as a graph $\mathcal{G}$ (details of the encoding discussed in "Methods") and use a graph neural network $G$ to learn a representation for the corresponding polymer, $\boldsymbol{v}_{\mathcal{G}} = G(\mathcal{G})$. Here, we use a CGCNN[37] as $G$, similar to previous works that employ graph convolutional networks (GCNs) in polymers[38,39].

To build a predictor that reduces random errors, we use the robustness of neural networks against random noises in the training data, previously demonstrated in images[20] and graphs[21]. We assume that there exists a true target property (e.g., conductivity) that is uniquely determined by the structure of the polymer (which would require infinite repetitive simulations to obtain), and the computed target property from MD is slightly different from the true property due to the random errors in the simulation. This assumption can be written as,

$$t = f(\mathcal{G}) + \epsilon, \qquad (1)$$

where $t$ is the target property computed from MD, $f$ is a deterministic function mapping from monomer structure to true polymer property, and $\epsilon$ is a random variable independent of $\mathcal{G}$ with zero mean. Note that $\epsilon$ should be a function of $\mathcal{G}$ in principle, but similar noise is observed across polymers as shown in Supplementary Fig. 2 and assuming $\epsilon$ is independent of $\mathcal{G}$ simplifies our analysis. By regressing over $t$, it is possible to learn $f(\mathcal{G})$ even when the noise is large[20] if enough training data is available. To generate a large amount of training data, since 50 ns simulations are too expensive practically, we use less accurate 5 ns simulations to generate training data and use a network $g_1$ to predict $t_5$ ns with the graph representation,

$$y_{5\mathrm{ns}} = g_1(\boldsymbol{v}_{\mathcal{G}}). \qquad (2)$$

With enough training data generated using the affordable 5 ns simulations, we can learn an approximation to the true property function $f_{5\,\mathrm{ns}}$ despite the random errors. However, there is a systematic error between $f_{5\,\mathrm{ns}}$ and $f_{50\,\mathrm{ns}}$ due to the slow relaxation of polymers. To correct this error, we perform a small amount of 50 ns simulation to generate data for the converged conductivities. This correction can then be learned with a linear layer $g_2$ using both predictions from 5 ns simulations and the graph representations,

$$y_{50\,\mathrm{ns}} = g_2(\boldsymbol{v}_{\mathcal{G}} \parallel y_{5\,\mathrm{ns}}), \qquad (3)$$

where $\parallel$ denotes concatenation.

Finally, the two datasets, a larger 5 ns dataset and a smaller 50 ns dataset, can be trained jointly using a combined loss

function,

$$\mathrm{Loss} = (1-w) \cdot \frac{1}{N_{5\,\mathrm{ns}}} \sum_{\mathcal{G}_{5\,\mathrm{ns}}} (y_{5\,\mathrm{ns}} - t_{5\,\mathrm{ns}})^2 + w \cdot \frac{1}{N_{50\,\mathrm{ns}}} \sum_{\mathcal{G}_{50\,\mathrm{ns}}} (y_{50\,\mathrm{ns}} - t_{50\,\mathrm{ns}})^2,$$

(4)

where $w$ is a weight between 0 and 1.

Using an iterative scheme, we sampled the entire polymer space in Fig. 1a with both 5 ns and 50 ns simulations. The 5 ns dataset includes 876 polymers and the 50 ns dataset includes 117 polymers. Note that we only simulate each polymer once so there is no duplicate in both datasets. We leave 10% of the polymers in both datasets as test data, and use tenfold cross-validation on the rest of the data to train our models. Due to the small size of the 50 ns dataset, we use stratified split while dividing the data to ensure that the training, validation, test data contain polymers with the full range of conductivities[40]. In the next sections, we first demonstrate the performance of our model based on these two datasets and then discuss the iterative screening of the polymer space.

**Performance on reducing random errors.** To demonstrate that our model can recover the true properties from noisy data, we first study a toy dataset for which we have access to the true property $f(\mathcal{G})$ in Eq. (1). We use the same dataset from 5 ns simulations and compute the partition coefficient, LogP, of each polymer using Crippen's approach[41,42], which uses an empirical equation whose output is fully determined by the molecular structure. Then, we add different levels of Gaussian random noise into the LogP values to imitate the random errors in simulated conductivities. Here, we only use the $g_1$ branch of our model, i.e., $w = 0$, to predict LogP values from the synthesized noisy data. Figure 2a shows the true mean absolute errors (MAEs) with respect to the original LogP values and apparent MAEs with respect to the noisy LogP values as a function of the standard deviation of the Gaussian noise, on a test dataset including 86 polymers. We observe that the true MAEs become smaller than the mean absolute deviation (MAD) of the Gaussian noise when the noise standard deviation is larger than 0.08. This result shows that our model predicts LogP more accurately than performing a noisy simulation of LogP due to the existence of large random error in the simulation. The random error reduction is possible because structurally similar polymers tend to have similar properties. Since the random errors in each MD simulation is independent, the random fluctuations in the simulated properties will cancel out for structurally similar polymers during the training of the GCN.

We cannot use the same approach to evaluate the model performance on predicting simulated 5 ns conductivities because we do not have access to the true conductivities. Therefore, we make an approximate evaluation by running another independent MD simulation for each test polymer and compare our predicted conductivity to the mean conductivity from the two independent simulations, i.e., the original simulation (config A) and the new simulation (config B). In Fig. 2b, the MAE on 86 test data is 0.078 $\log_{10}$(S/cm), which is smaller than the corresponding random error from simulation of 0.094 $\log_{10}$(S/cm) (computed by the MAE between the two independent MD simulations in Fig. 2c divided by $\sqrt{2}$). This result indicates that our prediction of the noisy conductivity also outperforms an independent MD simulation due to its large random noise, similar to the LogP prediction. In Supplementary Fig. 3, we employ a random forest (RF) model with the Morgan fingerprint[43] of the polymer structure to predict the conductivity, achieving an MAE of 0.099 $\log_{10}$(S/cm). This result shows that RF has slightly worse performance than GNN, causing the errors to be larger than the random errors in the simulated conductivities. To estimate the true prediction performance with respect to the inaccessible true conductivity, we need to assume that the random errors for 5 ns MD conductivity follow a Gaussian distribution, which is approximately correct (Supplementary Fig. 2). We could then estimate the true root mean squared error (RMSE) to be 0.060 $\log_{10}$(S/cm), smaller than the standard deviation of the Gaussian noise 0.117 $\log_{10}$(S/cm). Further, we estimate that our GNN prediction accuracy is the accuracy of running ~4 MD simulations for each polymer (detailed calculations can be found in Supplementary Note 1).

**Performance in correcting systematic errors.** In addition to reducing random errors, our model is also able to learn the systematic difference between 5 ns and 50 ns MD simulated properties with the multitask scheme. After co-training our model with both 5 ns and 50 ns datasets, we present the predictions on 11 test data from 50 ns MD in Fig. 3a. Compared with the original 5 ns conductivities, our model corrects the systematic error and achieves a MAE of 0.076 $\log_{10}$(S/cm) by averaging the predictions from tenfold cross-validations. It is clear that the model corrects the systematic error by learning a customized correction to each polymer, which is better than an overall linear correction which

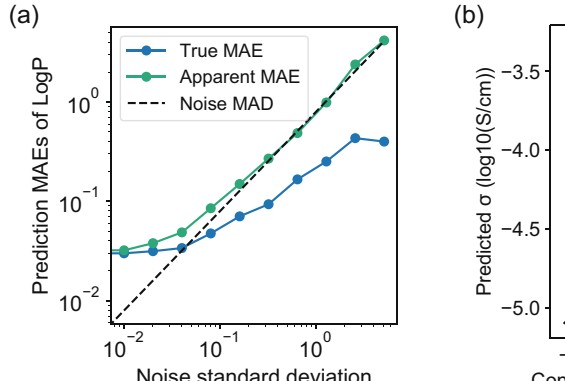
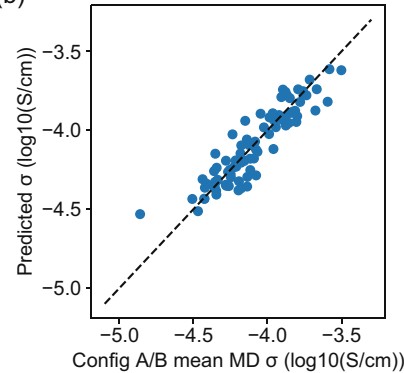
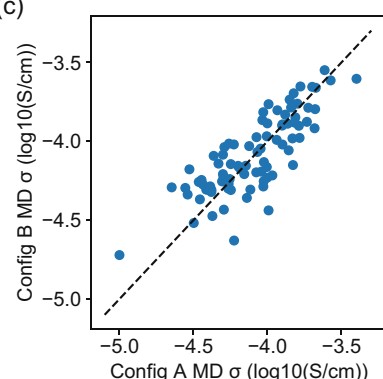

**Fig. 2 Performance on reducing random errors. a** Mean absolute errors (MAEs) on a toy dataset to predict LogP with increasing noises in training data. Blue line denotes MAEs with respect to true LogP values, green line denotes MAEs with respect to noisy LogP values, and dashed line denotes the mean absolute deviation (MAD) of the Gaussian noise. **b** Scatter plot comparing the predicted conductivity and computed mean conductivity from two independent initializations (config A and config B) in the test dataset. **c** Scatter plot comparing conductivities from two independent initializations for the same polymers in the test dataset.

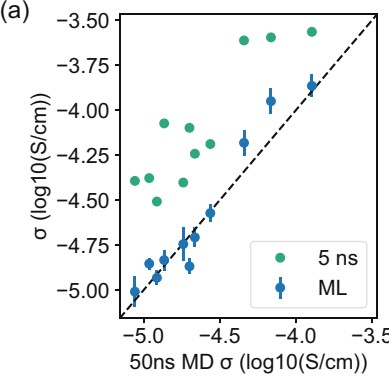
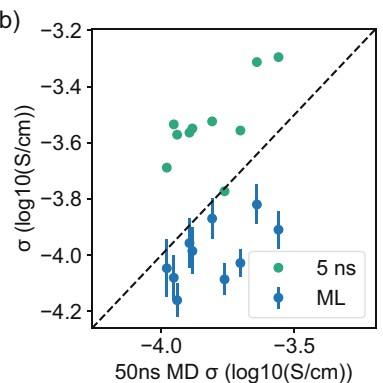
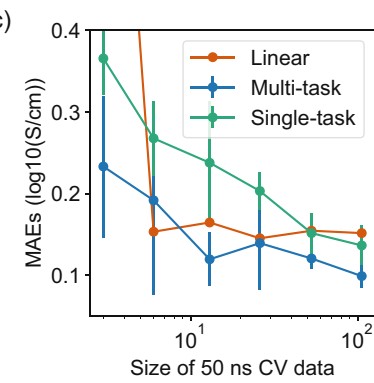

**Fig. 3 Performance on correcting systematic errors. a, b** Scatter plots showing the interpolation (**a**) and extrapolation (**b**) performance of the model on test data. Blue and green dots present the results of 5 ns MD simulations and ML predictions compared with 50 ns MD conductivities, respectively. The error bars represent the standard deviations of predictions from 10-fold cross-validation. **c** Change of interpolation performance with the different numbers of CV data. The red, blue, green lines denote the MAEs of linear correction, multitask model, and single-task model predicting 50 ns conductivity. The error bars represent the standard deviations of MAEs from tenfold cross-validation.

gives a MAE of 0.152 $\log_{10}$(S/cm). Note that this MAE does not include random errors, because our 5 ns and 50 ns conductivities are computed from the same random initial structures. The results in Fig. 3a represent the interpolation performance of our model since we randomly split our data. To further study the extrapolation performance, we perform the same co-training but reserve the top ten polymers with the highest conductivity as test data. In Fig. 3b, we find that by training with low-conductivity polymers, the model underestimates the 50 ns conductivity and achieves a MAE of 0.182 $\log_{10}$(S/cm). This underestimation is due to the larger systematic error between 50 and 5 ns conductivities in training data, caused by slow relaxations in low-conductivity polymers and the possible different transport mechanism between low- and high-conductivity polymers. Nevertheless, the model still performs better than a linear correction that only has access to the training data, which has a MAE of 0.275 $\log_{10}$(S/cm).

In Table 1, we study how the systematic error correction performs for other transport properties, including lithium-ion diffusivity ($D_{Li}$), TFSI diffusivity ($D_{TFSI}$), and polymer diffusivity ($D_{Poly}$). Both interpolation and extrapolation performances are reported similar to the results of conductivity. To better evaluate the uncertainties caused by the small 50 ns dataset, we compute the mean and standard deviation of the prediction MAEs from each fold of tenfold cross-validation in *GCN CV*. This MAE is different from our previous MAEs, denoted as GCN average, which uses the mean from cross-validations to make a single prediction. Overall, ML average outperforms a linear correction for all properties, indicating the generality of the customized correction of systematic errors. However, there is a relatively high variance between different folds of cross-validation due to the small data size, especially for the extrapolation tasks. *GCN CV* performs the same or slightly worse than a linear correction for $D_{TFSI}$, $D_{Poly}$, and $D^*_{Poly}$. A potential explanation is that a linear correction already performs reasonably well for these properties, demonstrated by the small MAEs of linear correction, while a more complicated multitask model is prone to overfitting the noises in a small 50 ns dataset. Due to the relative small size of our training data, we develop a simpler multitask random forest (RF) model that mimics the multitask GCN architecture in Fig. 1d (details described in Supplementary Note 2). However, the RF model performs worse than GCN in all properties as shown in Supplementary Table 1, which is consistent with the relative poor performance of RF in random error reduction.

In Fig. 3c, we further study how the performance of our model would evolve with less 50 ns data, since these long MD simulations

are expensive to run and cannot be easily parallelized. We find that the performance of the multitask model decreases relatively slowly with less training data, and it still has some correction ability even with 13 CV data points, despite the large uncertainties due to the small data size. This observation shows the advantage of co-training a larger 5 ns dataset and a smaller 50 ns dataset—it is much easier to learn a systematic correction than learn the property from scratch, and the co-training allows the transfer of graph representation learning from the 5 ns dataset to the 50 ns dataset. In contrast, the performance of a single-task model directly predicting 50 ns conductivity degrades much faster with less training data.

**Acceleration of the screening of polymers.** After demonstrating the performance of the multitask model on reducing both random and systematic errors, we employ this model to perform an extensive screening of polymer electrolytes in the polymer space defined in Fig. 1a. The goal of the screening is to search for polymers with the highest conductivity. As shown in Fig. 4a, we obtain 53,362 polymer candidates using polymerization criteria from the ZINC chemical database[26]. To reduce the average computational cost, we limit our search space to only include polymers with monomer molecular weight less than 200, resulting in 6247 polymers. As shown in Supplementary Figs. 6 and 7, both search and candidate spaces cover a diverse set of polymer structures.

We first use 5 ns MD simulations and a single-task GCN to explore polymers in the search space. To reduce the computational cost, we only simulate each polymer once and employ GCN to reduce the random errors in the simulation. We perform 300 simulations in each iteration, 150 on randomly sampled polymers and 150 on best polymers predicted by GCN, which balances the exploration and exploitation. As shown in Fig. 4b, the conductivities of the top 50 polymers gradually increase as more polymers are explored with the iterative approach. But after 900 simulations, the average conductivity only increases slightly, indicating that we have achieved the best polymers in the 6247 search space based on 5 ns simulations.

Due to the systematic differences between 5 and 50 ns simulations, we randomly sample 120 polymers from those 900 polymers (876 successful simulations) and perform additional 50 ns MD, in which 117 are successful. These data allow us to correct the systematic errors in 5 ns simulation using the multitask model. We note that in previous sections we already use some data from the screening workflow to demonstrate the model performance. In Fig. 4c, we use the multitask model to

**Table 1 Comparison of the mean absolute errors (MAEs) on predicting 50 ns MD simulated properties between different approaches.**

| Method | $\sigma$ | $\sigma^*$ | $D_{Li}$ | $D_{Li}^*$ | $D_{TFSI}$ | $D_{TFSI}^*$ | $D_{Poly}$ | $D_{Poly}^*$ |
|---|---|---|---|---|---|---|---|---|
| 5 ns (direct) | 0.528 | 0.278 | 0.503 | 0.419 | 0.455 | 0.249 | 0.612 | 0.528 |
| 5 ns (linear) | 0.152 | 0.275 | 0.148 | 0.247 | 0.096 | 0.297 | 0.072 | 0.110 |
| GCN CV | 0.093 ± 0.017 | 0.186 ± 0.053 | 0.106 ± 0.016 | 0.209 ± 0.050 | 0.101 ± 0.020 | 0.181 ± 0.028 | 0.072 ± 0.019 | 0.114 ± 0.030 |
| GCN average | 0.076 | 0.182 | 0.080 | 0.202 | 0.075 | 0.171 | 0.056 | 0.104 |

The first row denotes the MAE between 5 and 50 ns simulated properties. For each property, interpolation and extrapolation performance are represented by labels without and with the * symbol. Uncertainties are the standard deviations of MAEs from tenfold cross-validation (CV).

predict the 50 and 5 ns conductivities of all 6247 polymers in the search space. As a result of the customized correction, the ordering of conductivity changes from 5 to 50 ns predictions. The Spearman's rank correlation coefficient between these two predictions is 0.852, indicating that the ordering change is small but significant. For the top 50 polymers from 5 ns predictions, only 37 remain in the top 50 based on 50 ns predictions. This ordering change shows that the correction of systematic errors help us to identify some polymers that might be disregarded if only 5 ns simulations are performed.

To estimate the amount of acceleration we achieve, we compare the actual CPU hours used to the CPU hours that would be required if we performed one 50 ns MD simulations for each polymer. These simulations are run on NERSC Cori Haswell Compute Nodes and the CPU hours are estimated by averaging 100 simulations. In total, we use approximately 394,000 CPU hours for the MD simulations, with 33.2% for sampling and relaxing amorphous structure, 28.6% for 5 ns MD, and 38.2% for 50 ns MD. The total cost only accounts for around 4.4% and 0.51% of the computation needed to simulate all the polymers from the 6247 search space and the 53,362 candidates, respectively. Note that this conservative estimation assumes that only one 50 ns MD simulation is performed for each polymer for the brute-force screening. As shown in the previous section, our model has a true prediction error smaller than the random error from a 5 ns MD simulation. Although the random error from 50 ns simulation might be smaller, our model may have a larger acceleration due to the effect of random error reduction.

**Validation of the best candidates from the screening.** We employ the learned multitask model to screen all 6247 polymers in the search space and 53,362 polymers in the candidate space. In Fig. 5a, we use 50 ns MD to simulate ten polymers out of the top 20 in the search space and 14 polymers out of the top 50 in the candidate space. These polymers are randomly selected from the top polymers using Butina clustering[42,44] to reduce their structural similarity, and only polymers which have not been seen in the 50 ns dataset are selected. We observe a MAE of 0.120 $\log_{10}$(S/cm) and 0.093 $\log_{10}$(S/cm) for the predictions in search space and candidate space, respectively, which are between the interpolation and extrapolation errors in Fig. 3 and Table 1. It shows that the extrapolation to the candidate space is easier than our hypothetical extrapolation test in Fig. 3b, yet a similar underestimation of conductivity is observed in the extrapolation. The larger errors for the top polymers in the search space might be explained by a combination of extrapolation errors and random errors in 50 ns MD simulations. We summarize the structure of the top polymers in Supplementary Tables 2 and 3, and most of them have PEO-like substructures which might explain their relatively high conductivity.

In Fig. 5b, we further validate the prediction of the model by gathering experimental conductivities for 31 different polymers from the literature which are measured at the same salt concentration and temperature as our simulations[4,6,27–36], and the

results are also summarized in Supplementary Table 4. Note that some polymers, like polyethylene oxide (PEO), do not follow the same structural pattern as our polymers. Nevertheless, the model still gives a reasonable prediction on these out-of-distribution polymers because there are many PEO-like polymers in the training data. The largest errors come from the polymers with experimental conductivity less than $10^{-5}$ S/cm. In general, it is difficult to simulate the conductivity of polymers with such low conductivity due to the long MD simulation time needed for convergence. In Supplementary Fig. 4, we observe a much smaller prediction error with respect to 50 ns MD simulated conductivities for these polymers, indicating that the error with respect to the experiments is likely caused by the limited simulation time in MD. Other than the difficulty of simulating low-conductivity polymers, possible causes of the error also include the inaccuracy of the force fields, the finite length of the polymer chain, the finite size of the simulation box, etc. For the top polymers like PEO, we observe an underestimation of conductivity because the model cannot extrapolate to these polymers that are significantly different from the training data. It is also possible to incorporate the experimental data in our multitask GCN model to correct this simulation error with respect to experiments. In Supplementary Fig. 5, we show the predicted experimental conductivities by replacing the 50 ns MD data with experimental data in the multitask GCN. However, due to the limited size of experimental data, it is challenging to evaluate the predictions without further experiments.

**Insights for polymer electrolyte design.** The polymer electrolyte space screened in this study is significantly larger than previous works, and it contains less human bias because the candidates are randomly sampled from large databases. Therefore, we can draw more statistically meaningful conclusions to some important questions for polymer electrolyte design. In Fig. 6a, we find that there is an optimum ratio of solvating sites of around 0.4, approximated by the atomic percentage of N, O, S atoms to non-hydrogen heavy atoms, to maximize Li-ion conductivity. A previous study indicates that higher solvation-site connectivity leads to a higher conductivity for PEO-like polymers[27], whose maximum oxygen percentage is 0.33 for PEO. Our results indicate that an even higher ratio of solvating sites might harm conductivity due to increased glass transition temperature from strong solvating site interactions[45,46]. In Fig. 6b, we observe that introducing side chains to the polymer backbone decreases the Li-ion conductivity, which might be explained by the difficulty of forming solvation sites with side chains compared with a simple linear chain. We note that general statistical correlations may not apply to carefully designed structural modifications to individual polymers. For instance, previous studies have shown that introducing ethyleneoxy (EO) side chains can improve the conductivity of polymer electrolytes[47].

We further explore the atomic-scale mechanisms that limit the conductivity in polymer electrolytes. A well-known hypothesis is

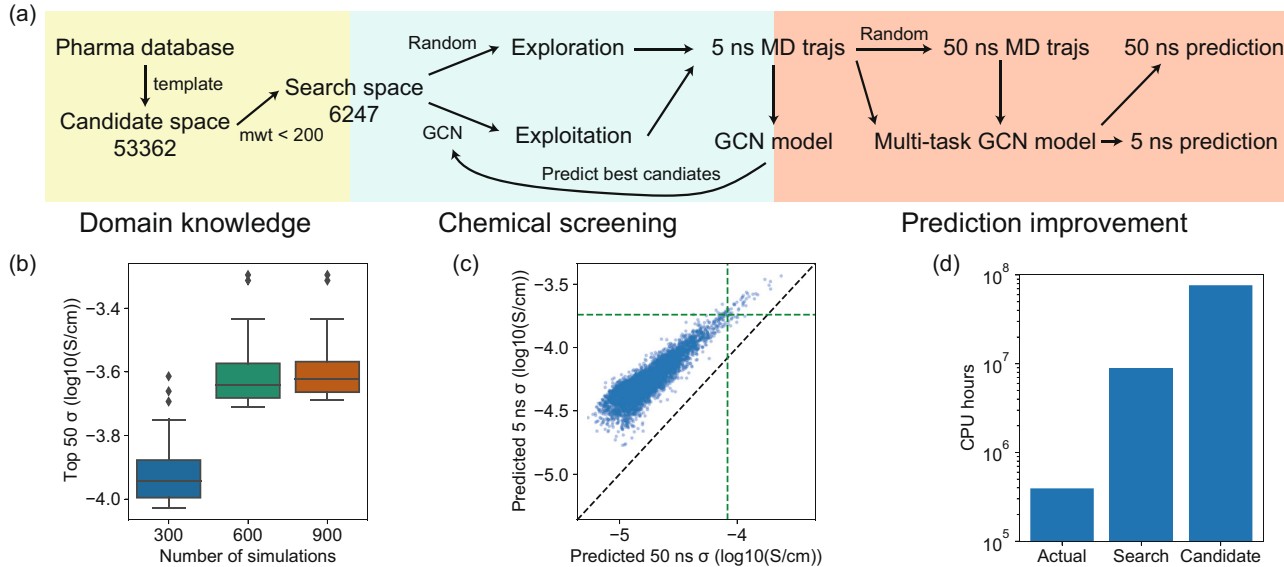

**Fig. 4 Screening of polymer electrolytes. a** Illustration of the screening workflow. **b** Distribution of the conductivities of top 50 polymers after each iteration, showing the quartiles of conductivity distributions. **c** Predictions of 50 ns and 5 ns conductivities for 6247 polymers in the search space. Green line denotes the top 50 conductivity from both predictions. **d** CPU hours that are actually used, required to screen the entire 6247 search space, and required to screen the 53362 candidate space.

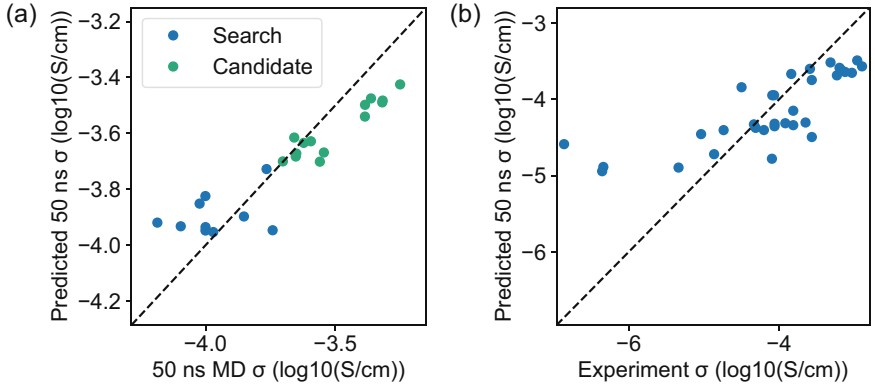

**Fig. 5 Validation of the predicted polymers. a** Validation of the best candidates from the search space (blue) and the candidate space (green). **b** Validation of the model prediction with out-of-distribution literature data.

that Li-ions transport in polymers via segmental motion mechanism, rather than the ion hopping mechanism in ceramic solid electrolytes[1,48]. We examine this hypothesis by computing the ratio between predicted Li-ion diffusivity and polymer diffusivity. In Fig. 6c, this ratio is between 0.59 and 3.63 for all polymers, while most high-conductivity polymers have this ratio below 1. This result supports the segmental motion hypothesis because the Li-ion and polymer dynamics are strongly coupled, at least for high-conductivity polymers. The lack of polymers in the upper right of the plot indicates none of the high-conductivity polymers employs an ion hopping mechanism. Therefore, the exploration of such polymers requires a chemical structure far different from our search space. We believe more scientific insights can be obtained from our data, therefore we provide all four predicted 50 ns MD properties for 6247 polymers in the search space and 53,362 polymers in the candidate spaces in the supplementary materials for the community.

## Discussion

We have performed a large-scale computational screening of polymer electrolytes by learning to reduce random and systematic errors from molecular dynamics simulation with a multitask learning framework. Our screening shows that the PEO-like structure is the optimum structure for a broad class of carbonyl-based polymers. Although the result may seem unsurprising because PEO has been one of the best polymer electrolytes since its discovery in 1973[49], it shows the advantage of PEO-like polymers over a very diverse set of chemical structures. The only constraint of the polymer candidates is to have a carbonyl structure, and the rest of the structure is randomly sampled from a large database of drug-like molecules[26], containing few human biases. Since the PEO substructure automatically emerge from the candidates, it indicates that the PEO substructure has an advantage over almost all other types of chemical structures in the diverse database, given the existence of a carbonyl group in the polymer. This result might explain why PEO is still one of the best polymer electrolytes despite a significant effort to find better candidates in the community. Several potential directions remain open for discovering polymer electrolytes better than PEO. The first is to search for polymer electrolytes that achieve optimum conductivity at very high salt concentrations. Conductivity generally increases with increased salt concentration, but ion clustering and decreased diffusivity will reduce conductivity at high

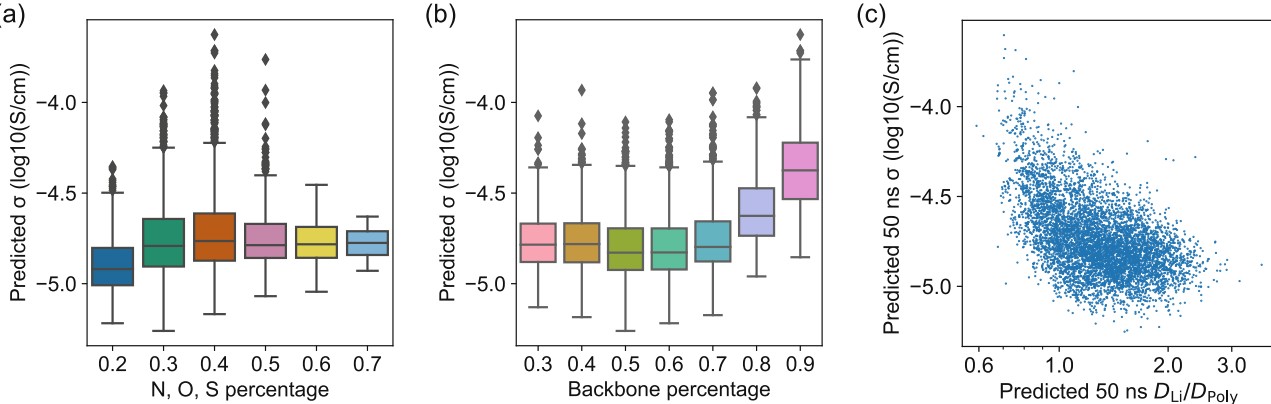

**Fig. 6 Relation between several descriptors and predicted 50 ns MD Li-ion conductivity for polymers in the 6247 search space. a** The percentage of N, O, S atoms to non-hydrogen heavy atoms in the polymer structure. **b** The percentage of backbone atoms to non-hydrogen heavy atoms in the polymer structure. **c** The ratio between predicted Li-ion and polymer diffusivity, corresponding to the degree of decoupling between Li ion and polymer dynamics.

concentrations[1]. Our screening keeps a constant concentration of 1.5 mol/kg LiTFSI for different polymers, but some polycarbonate electrolytes show advantage at an extremely high salt concentrations[50,51]. The second is to explore polymer chemistry beyond this study. Due to the limitations of the Monte Carlo procedure used to generate initial configurations, our simulations do not include polymers with aromatic rings. Recent studies propose the potential of polymers with high fragility and aromatic rings as polymer electrolytes due to the decoupling of ionic conductivity from structural relaxation[52]. Backbones containing different lewis acidic heteroatoms or non-carbonyl-based motifs could also lead to better polymer electrolytes[9].

The large-scale screening is possible because we significantly reduce the computational cost of individual simulations by learning from imperfect data with the multitask learning framework. The ability of neural networks to learn from noisy data is extensively studied in machine learning[20,53,54] and has recently been applied to reduce the signal-to-noise ratio of band-excitation piezoresponse force microscopy[55] in materials science. Despite the wide use of graph neural networks in material discovery[18,56,57], the random errors in training data are less studied, possibly because previous studies focus on simpler materials of which the random errors are much smaller. We show that random errors can be effectively reduced by learning a graph neural network across different chemistry even when the random error for each simulation is significant. It provides a potentially generalizable approach to accelerate the screening of complex materials whose structures can only be sampled from a distribution, e.g., amorphous polymers, surface defects, etc., because only one, instead of several, simulation needs to be performed for each material by adopting our approach.

The systematic error reduction demonstrated in this work is closely related to the transfer learning studies that aim to combine data from different sources[22,24,58,59]. Our unique contribution in this work is to demonstrate the value of short, unconverged MD simulations in the context of material screening. We find that the systematic error between the 5 and 50 ns simulated transport properties can be corrected with a small amount of 50 ns simulations, which can potentially be generalized to other types of materials, properties, and simulation methods. Because our multitask GCN architecture uses the 5 ns properties as an additional input to predict 50 ns properties, it is also conceptually similar to the delta-learning approach[60]. In summary, we hope that the random and systematic error reductions observed in this work could highlight the value of imperfect, cheaper simulations for material screening that might previously be overlooked. A

broader class of complex materials could be screened with a similar approach if a cheap, noisy, and biased simulation method can be identified.

## Methods

**Graph representation for polymers**. The polymers are represented by graphs based on their monomer structure. The node embeddings $\boldsymbol{v}_i$ and edge embeddings $\boldsymbol{u}_{ij}$ are initialized using atom and bond features described in Supplementary Tables 5 and 6. An additional edge is added to connect two ends of the monomer, allowing the end atoms to know the local chemical environments. We find that this representation has a better performance than using dummy atoms to denote the monomer ends.

**Network architecture**. We employ a graph convolution function developed in ref. [37] to learn the node embeddings in the graph. For each node $i$, we first concatenate the center node, neighbor, and edge embeddings from last iteration $\boldsymbol{z}_{(i,j)}^{(t-1)} = \boldsymbol{v}_i^{(t-1)} \parallel \boldsymbol{v}_j^{(t-1)} \parallel \boldsymbol{u}_{(i,j)}$, then perform graph convolution,

$$\boldsymbol{v}_i^{(t)} = \boldsymbol{v}_i^{(t-1)} + \sum_{j \in Neigh(i)} \sigma(\boldsymbol{z}_{(i,j)}^{(t-1)} \boldsymbol{W}_f^{(t-1)} + \boldsymbol{b}_f^{(t-1)}) \cdot g(\boldsymbol{z}_{(i,j)}^{(t-1)} \boldsymbol{W}_s^{(t-1)} + \boldsymbol{b}_s^{(t-1)}), \quad (5)$$

where $\boldsymbol{W}_f^{(t-1)}$, $\boldsymbol{W}_s^{(t-1)}$, $\boldsymbol{b}_f^{(t-1)}$, $\boldsymbol{b}_s^{(t-1)}$ are weights, $\sigma$ and $g$ are sigmoid and softplus functions, respectively. After learning the node embeddings, we use a global soft-attention pooling developed in ref. [61] to learn a graph embeding,

$$\boldsymbol{v}_\mathcal{G} = \sum_i \text{softmax}(h_{\text{gate}}(\boldsymbol{v}_i)) \cdot h(\boldsymbol{v}_i), \quad (6)$$

where $h_{\text{gate}} : \mathbb{R}^F \to \mathbb{R}$ and $h : \mathbb{R}^F \to \mathbb{R}^F$ are two fully connected neural networks. The graph embedding $\boldsymbol{v}_\mathcal{G}$ is then used in Eq. (2) and Eq. (3) to predict polymer properties.

**Molecular dynamics simulations**. The molecular dynamics simulations are performed with the large atomic molecular massively parallel simulator (LAMMPS)[62]. The atomic interactions are described by the polymer consistent force field (PCFF +)[63,64], which has been previously used for polymer electrolyte systems[10,13,65]. The charge distribution of TFSI$^-$ is adjusted following ref. [66], using a charge scaling factor of 0.7, to better describe the ion-ion interactions. All partial charges are reported in Supplementary Table 7. There are 50 Li$^+$ and TFSI$^-$ in the simulation box. Each polymer chain has 150 atoms in the backbone. The number of polymer chains is determined by fixing the molality of LiTFSI at 1.5 mol/kg. The initial configurations are generated using a Monte Carlo algorithm, implemented in the MedeA simulation environment[67]. The 5-ns-long equilibration procedure is based on a scheme described in ref. [13]. Once equilibrated, the system is then run in the canonical ensemble (nVT) at a temperature of 353 K, using a rRESPA multi-timescale integrator[68] with an outer timestep of 2 fs for nonbonded interactions, and an inner timestep of 0.5 fs. The high-throughput workflow is implemented using the FireWorks workflow system[69]. To resolve unexpected errors during MD simulations, the workflow will try to restart the simulation three times and disregard the simulation if all three simulations are failed.

**Calculation of transport properties**. The diffusivities of lithium and TFSI ions are calculated using the mean squared displacement (MSD) of the corresponding

particles,

$$D = \frac{\left\langle [\boldsymbol{x}_i(t) - \boldsymbol{x}_i(0)]^2 \right\rangle}{6t}, \tag{7}$$

where $\boldsymbol{x}$ is the position of the particle, $t$ is the simulation time, and $\langle \cdot \rangle$ denotes an ensemble average over the particles. The diffusivity of the polymer is calculated by averaging the diffusivities of O, N, and S atoms in the polymer chains. The conductivity of the entire polymer electrolyte is calculated using the cluster Nernst-Einstein approach developed in ref. [65]. This method takes into account ion-ion interactions in the form of aggregation of ion clusters,

$$\sigma = \frac{e^2}{V k_B T} \sum_{i=0}^{N_+} \sum_{j=0}^{N_-} z_{ij}^2 \alpha_{ij} D_{ij}, \tag{8}$$

where $\alpha_{ij}$ is the population of the ion clusters containing $i$ cations and $j$ anions, $z_{ij}$, $D_{ij}$ are the charge and diffusivity of the cluster, $N_+$ and $N_-$ are the maximum number of cations and anions in the clusters, $e$ is the elementary charge, $k_B$ is the Boltzmann constant, and $V$ and $T$ are the volume and the temperature of the system. We use the $cNE_0$ approximation that assumes $D_{ij}$ is equal to the average diffusivity of lithium ion if the cluster is positively charged, and TFSI ion if the cluster is negatively charged[65].

## Data availability

The toy LogP dataset, the 5 ns, and 50 ns MD datasets are available in Supplementary Data 1. The CGN predicted 50 ns conductivity, Li-ion diffusivity, TFSI diffusivity, and polymer diffusivity for the 6247 search space and 53,362 candidate space are available in Supplementary Data 1. The experimentally measured conductivity from literature is available in Supplementary Table 4. The raw MD trajectories are too large to be shared publicly. We are developing a database to facilitate the sharing and they will be made available in the future.

## Code availability

The multitask graph neural network is implemented with PyTorch[70] and PyTorch Geometric[71]. The code is available at https://github.com/txie-93/polymernet.

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

## Acknowledgements

This work was supported by Toyota Research Institute. Computational support was provided by the National Energy Research Scientific Computing Center, a DOE Office of Science User Facility supported by the Office of Science of the U.S. Department of Energy under Contract No. DE-AC02-05CH11231, and the Extreme Science and Engineering Discovery Environment, supported by National Science Foundation grant number ACI-1053575. J.L. acknowledges support by an appointment to the Intelligence Community Postdoctoral Research Fellowship Program at the Massachusetts Institute of Technology, administered by Oak Ridge Institute for Science and Education through an interagency agreement between the U.S. Department of Energy and the Office of the Director of National Intelligence.

## Author contributions

T.X. developed the machine learning algorithm. T.X., A.F.-L., and Y.W. designed and performed the molecular dynamics simulation. T.X., M.A.S., and M.H. designed the polymer candidate space. J.L. gathered the data from the literature. T.X., J.C.G., Y.S.H., J.A.J., and R.G.B. conceived the idea and approach. T.X., A.F.-L., Y.W., J.L., M.A.S., M.H., G.M.L., R.G.-B., J.A.J., Y.S.-H., and J.C.G. contributed to the interpretation of the results and the writing of the paper.

## Competing interests

The authors declare no competing interests.
