## [Peer Review File · Nature Communications]

REVIEWER COMMENTS

Reviewer #1 (Remarks to the Author):

The major bottleneck in the calculation of polymer properties using MD simulation is the long computation time required for equilibration. The authors introduced a multi-task deep learning technique to reduce the computational cost. In addition to a small number of long-run MD data, which are computationally expensive, the authors utilized a large number of short-run MD datasets including systematic bias to be removed. By estimating the systematic bias between these two datasets, the results of short-run MD (source domain) were corrected to the properties of long-run MD (target domain). This domain shift was done in the framework of multi-task learning.

This study deals with a very important problem in polymer property calculations. The results are also appealing. However, I think that the scientific results obtained do not meet the standards required by Nature Communications.

1. First of all, the scope of the paper needs to be clarified. I understand that the main result of this research lies in the proposal of a new methodology because the present method has no special idea and physical insights about the ionic conduction of polymer electrolytes. However, the title of the paper contains the words "amorphous polymer electrolytes", and the empirical experiments are only on the ionic conductivity of polymer electrolytes (with several related properties). Is this paper intending to propose a methodology for this limited system or a general methodology applicable to a broader system? Please clarify the position of this research and the scope of application of the methodology.

2. On the other hand, there is no in-depth discussion on physicochemical aspects of the identified candidates for polymer electrolytes. There has been no progress in experiments or theory. If the authors are going to focus on "amorphous polymer electrolytes", at least some scientific contribution to physics or chemistry is required.

3. From the comparison of Figs. 2(b) and (c), the authors argue that in building a surrogate model of machine learning, it is not necessary to perform repeated calculations for the same polymer, but learning the results of single calculations for multiple polymers can result in a reduction in the variance of MD simulations, even without repeated calculations of the same polymer. There is a logical contradiction in this claim. Here the variance of the MD iterations is denoted by σ , then the estimated variance of the sample mean calculated from n iterations is σ/n . The authors should compare the estimated variance of the MD simulation, σ/n , and the MAE (or its squared scale) of the machine learning model, but the comparison between Fig 2(b) and Fig 2(c) only compares σ and MAE.

The proposed method addresses the statistical problem of systematic bias, but it does not seem to have an error reduction mechanism. I understand what the authors are trying to claim, but the way of demonstration and argumentation needs to be improved. At least the term "error reduction" is misleading.

4. The present multi-task GCN is also a deterministic model, although it is referred to as "most previous studies implicitly assume that the properties used to train the ML models are generated through a deterministic, unbiased process" on page 3.

5. Did the 5ns and 50ns data sets contain the same polymers? Did the authors select polymers completely at random? It would be advantageous to include 5ns and 50ns data for the same polymer, at least in part, to estimate the systematic bias between the two. Please describe the analysis procedure in more detail. The framework of design of experiments, such as active learning or Bayesian optimization, should be introduced to rationally select effective computational targets for bias correction.

6. I could not find a detailed description of the MD simulation. The procedure needs to be detailed enough to reproduce the results of the paper, including the force field parameters, generation of the initial structure, conditions for the equilibration calculations, error checking, and software used.

Reviewer #2 (Remarks to the Author):

The authors eagerly addressed the reviewers' concerns.

Especially, I'm happy to see the additional data of a) random forest regression, b) example structures of 53k candidates, and c) comparison with experimental values. The discussion became more solid.

Since the work seems a kind of SOTA integration of computational approaches, the paper can be published in this journal.

I hope the authors will show as much code and data as possible so that the readers can trace the results.

(I couldn't access the code, as the text "The code is available at URL" on p.20 did not give available URL during the review)

Reviewer #3 (Remarks to the Author):

The manuscript by T. Xie et al. is a very interesting and well written piece of work and potentially worthy of being published in Nature Communications. The authors have responded well to the previous reviewers' comments and addressed nearly all the concerns satisfactorily. The flow of the paper has been improved based on the corrections made. I recommend the manuscript for publication, subject to answers to a few more questions that I list below.

1. The authors demonstrate with a toy problem that the GCN will have prediction errors smaller than random errors. However, do the authors have a good idea of how much random error they expect over several MD simulations performed for a given polymer starting from different configurations? From what I can see, comparisons have been made between configurations A and B but that does not provide an estimate of the random error.

2. I am not sure about the extrapolation claims made in the paper. Clearly the extrapolation errors are quite high, even if they perform better than linear correction. Normally, during machine learning exercises, one assumes the model cannot extrapolate and takes that into account in the training data. Please clarify/elaborate.

3. Has the multi-task GCN performance been compared to other traditional multi-fidelity learning approaches like delta learning, co-kriging, using the lower fidelity values as descriptor inputs, etc.?

Reviewers' Comments:

Reviewer #1 (Remarks to the Author):

The major bottleneck in the calculation of polymer properties using MD simulation is the long computation time required for equilibration. The authors introduced a multi-task deep learning technique to reduce the computational cost. In addition to a small number of long-run MD data, which are computationally expensive, the authors utilized a large number of short-run MD datasets including systematic bias to be removed. By estimating the systematic bias between these two datasets, the results of short-run MD (source domain) were corrected to the properties of long-run MD (target domain). This domain shift was done in the framework of multi-task learning.

This study deals with a very important problem in polymer property calculations. The results are also appealing. However, I think that the scientific results obtained do not meet the standards required by Nature Communications.

We thank the reviewer for the careful evaluation of our manuscript and helpful comments. The concerns are mainly focused on our scientific contributions. We would like to first highlight several key scientific contributions again:

- We performed the largest computational screening of polymer electrolytes, covering a significantly larger polymer space than past computational or experimental studies.
- In this unbiased exploration, PEO-like substructures automatically emerge as the best performing candidates, which is consistent with current consensus in the community. Our screening reaffirms the advantage of PEO-like substructures over a very diverse set of chemical substructures.
- We open-sourced the largest computational amorphous polymer dataset to date, including multiple properties like conductivity, Li-ion diffusivity, TFSI diffusivity, etc.
- In the section “Insights for polymer electrolyte design”, we extract multiple design principles for polymer electrolytes from our screening and newly added examination of the transport mechanisms in polymer electrolytes.
- We note that the polymer electrolyte space screened in our work is significantly larger than past studies and contains fewer human biases, so more statistically meaningful conclusions can be drawn for the design principles.

To address the reviewer’s comments, we have performed experiments to obtain additional scientific results and significantly revised our manuscript. Below, we respond to the reviewer’s comments point-by-point.

1. First of all, the scope of the paper needs to be clarified. I understand that the main result of this research lies in the proposal of a new methodology because the present method has no special idea and physical insights about the ionic conduction of polymer electrolytes. However, the title of the paper contains the words "amorphous polymer electrolytes", and the

empirical experiments are only on the ionic conductivity of polymer electrolytes (with several related properties). Is this paper intending to propose a methodology for this limited system or a general methodology applicable to a broader system? Please clarify the position of this research and the scope of application of the methodology.

We appreciate the reviewer's suggestion to clarify the scope of the paper. This paper intends to propose a new methodology for the computational screening of amorphous materials, which is applied to the problem of polymer electrolyte discovery and enables the largest screening of amorphous polymer to date. We extract many scientific insights and design principles for polymer electrolyte discovery. Although the methodology is only demonstrated in the polymer systems, we show that it achieves good performance on the toy system and 4 different properties for the polymer electrolytes. The challenge we face in these systems is rather common for other amorphous materials, so we believe our approach can be applied to many different problems.

To further clarify our scope, we have completely rewritten the abstract of our manuscript. We believe this new abstract will make our contributions clearer.

2. On the other hand, there is no in-depth discussion on physicochemical aspects of the identified candidates for polymer electrolytes. There has been no progress in experiments or theory. If the authors are going to focus on "amorphous polymer electrolytes", at least some scientific contribution to physics or chemistry is required.

As mentioned above, we discuss several physicochemical aspects of the polymer candidates in the section "Insights for polymer electrolyte design". To provide more scientific results, we have added a new plot Fig. 6(c) and a new paragraph in the same section. This new plot examines the well-known segmental motion hypothesis for Li-ion transport in polymer electrolytes in a much broader chemical space than past studies. We find that almost all high conductivity polymers have the ratios of Li-ion diffusivity to polymer diffusivity lower than 1, which supports the segmental motion hypothesis. It also indicates that none of the high-conductivity polymers employs an ion hopping mechanism, which would have ratios significantly larger than 1. Chemical structures far different from what is explored in this work should be considered to design polymer electrolytes with an ion hopping mechanism.

In addition, we provide all 4 predicted 50 ns MD properties for 6247 polymers in the search space and 53362 polymers in the candidate spaces in the supplementary materials. We believe the data will be useful for the community to extract more insights for polymer electrolyte design.

3. From the comparison of Figs. 2(b) and (c), the authors argue that in building a surrogate model of machine learning, it is not necessary to perform repeated calculations for the same polymer, but learning the results of single calculations for multiple polymers can result in a reduction in the variance of MD simulations, even without repeated calculations of the same polymer. There is a logical contradiction in this claim. Here the variance of the MD iterations

is denoted by σ , then the estimated variance of the sample mean calculated from n iterations is σ^2/n . The authors should compare the estimated variance of the MD simulation, σ^2/n , and the MAE (or its squared scale) of the machine learning model, but the comparison between Fig 2(b) and Fig 2(c) only compares σ and MAE.

This is a great point. The proper comparison should be between the mean error and σ/\sqrt{n} , not σ/n . We have revised our manuscript to compare our prediction MAE (0.078 log10(S/cm)) to the MAE between the two independent MD simulations divided by $\sqrt{2}$ (0.094 log10(S/cm)). Our model still shows a clear reduction in random errors.

In fact, we have discussed this problem in the later text that compares true RMSE and the standard deviation of the Gaussian noise. A more detailed discussion can be found in our supplemental note 1, where we show our ML prediction accuracy is approximately the accuracy of running ~ 4 MD simulations for each polymer.

The proposed method addresses the statistical problem of systematic bias, but it does not seem to have an error reduction mechanism. I understand what the authors are trying to claim, but the way of demonstration and argumentation needs to be improved. At least the term "error reduction" is misleading.

We suspect the error reduction mechanism is the systematic correlation between 5 ns and 50 ns simulated properties. For instance, if there is a constant difference between 5 ns and 50 ns simulated properties, a linear model can perfectly correct this error. However, our model outperforms a linear model by learning a customized correction to each polymer. We believe this is a reasonable error reduction mechanism and thus use the term "systematic error reduction".

4. The present multi-task GCN is also a deterministic model, although it is referred to as "most previous studies implicitly assume that the properties used to train the ML models are generated through a deterministic, unbiased process" on page 3.

The reviewer may have some misunderstanding here. Our model is deterministic, but it can handle data that is non-deterministic. Compared with past studies, we do not assume the properties used to train our model, i.e., the y in training data, are deterministic. The y is treated as a distribution with an expected value of y_{true} (note that y_{true} is inaccessible yet deterministic). Our GCN aims to predict y_{true} from a noisy training data and thus is a deterministic model.

5. Did the 5ns and 50ns data sets contain the same polymers? Did the authors select polymers completely at random? It would be advantageous to include 5ns and 50ns data for the same polymer, at least in part, to estimate the systematic bias between the two. Please describe the analysis procedure in more detail. The framework of design of experiments, such as active learning or Bayesian optimization, should be introduced to rationally select effective computational targets for bias correction.

Yes, the 50 ns dataset contains a subset of the 5 ns dataset. The polymers in the 5 ns dataset are selected via an iterative process that balances the exploration and exploitation. The polymers in the 50 ns dataset are selected at random from the 5 ns dataset. This procedure is discussed in detail in the second and third paragraphs of the section “Acceleration of the screening of polymers” as well as in Fig. 4(a). The systematic bias between the 5 ns and 50 ns datasets is shown in the first row of Table 1, where the MAE between 5 ns and 50 ns simulated properties are listed. To make it clearer, we revised Table 1 and added a sentence in the caption.

We like the reviewer’s idea to employ active learning or Bayesian optimization. However, due to the length and scope of this paper, we feel it makes most sense to leave this direction for future research.

6. I could not find a detailed description of the MD simulation. The procedure needs to be detailed enough to reproduce the results of the paper, including the force field parameters, generation of the initial structure, conditions for the equilibration calculations, error checking, and software used.

The detailed description of the MD simulation can be found in the methods section “Molecular dynamics simulations”. Here one can find details related to force field parameters, generation of the initial structure, conditions for the equilibration calculations, and software. We have added additional error checking information in the revised version.

Reviewer #2 (Remarks to the Author):

The authors eagerly addressed the reviewers' concerns.

Especially, I'm happy to see the additional data of a) random forest regression, b) example structures of 53k candidates, and c) comparison with experimental values. The discussion became more solid.

Since the work seems a kind of SOTA integration of computational approaches, the paper can be published in this journal.

I hope the authors will show as much code and data as possible so that the readers can trace the results.

(I couldn't access the code, as the text "The code is available at URL" on p.20 did not give available URL during the review)

We thank the reviewer for the positive feedback. The code and data are already included in the supplemental materials. We will upload the code to GitHub upon acceptance. In the revised manuscript, we have further included all CGN predicted 50 ns conductivity, Li-ion diffusivity, TFSI diffusivity, and polymer diffusivity for the 6247 search space and 53362 candidate space at the supplemental information.

Reviewer #3 (Remarks to the Author):

The manuscript by T. Xie et al. is a very interesting and well written piece of work and potentially worthy of being published in Nature Communications. The authors have responded well to the previous reviewers' comments and addressed nearly all the concerns satisfactorily. The flow of the paper has been improved based on the corrections made. I recommend the manuscript for publication, subject to answers to a few more questions that I list below.

We appreciate the positive feedbacks from the reviewer. We provide point-to-point responses to the raised questions below.

1. The authors demonstrate with a toy problem that the GCN will have prediction errors smaller than random errors. However, do the authors have a good idea of how much random error they expect over several MD simulations performed for a given polymer starting from different configurations? From what I can see, comparisons have been made between configurations A and B but that does not provide an estimate of the random error.

We used two different ways to estimate the random errors: 1) we run 6 MD simulation with different random initialization for the same polymer (monomer SMILES CCC(CNCCN(C)CCOC(=O)[Au])O[Cu], [Au] and [Cu] denotes two connecting ends). The standard deviation is $0.094 \log_{10}(S/cm)$; 2) we run 2 MD simulation with different random initialization for 86 test polymers in Fig. 2(c). The standard deviation is $0.117 \log_{10}(S/cm)$, assuming the same noise distribution for all polymers (detailed derivation in supplementary note 1).

It is too expensive to run more than 2 MD simulations for many polymers. Since 1) and 2) gives similar errors, we think they give a reasonable estimate of the random error.

2. I am not sure about the extrapolation claims made in the paper. Clearly the extrapolation errors are quite high, even if they perform better than linear correction. Normally, during machine learning exercises, one assumes the model cannot extrapolate and takes that into account in the training data. Please clarify/elaborate.

This is a great point. We agree with the reviewer that extrapolation is difficult. However, since our model learns the systematic difference between 5 ns and 50 ns MD simulated properties, can this *difference* be extrapolated? This is the motivation behind our study of the extrapolation errors. The extrapolation errors are indeed high, but we think it is also surprising that our model can outperform a linear model. We believe our results indicate that it is much easier to learn the systematic error correction than the direct prediction of properties.

3. Has the multi-task GCN performance been compared to other traditional multi-fidelity learning approaches like delta learning, co-kriging, using the lower fidelity values as descriptor inputs, etc.?

Yes, during the development of our model, we compared several approaches including delta learning and using lower fidelity values as descriptor inputs. It seems non-trivial to incorporate co-kriging into the GCN architecture. Including lower fidelity values as descriptor inputs significantly improves the model performance but predicting y_{50ns} has a similar performance as predicting $(y_{50ns} - y_{5ns})$. This result motivated the use of our current architecture, which uses the predicted y_{5ns} as descriptor inputs to directly predict y_{50ns} .

REVIEWERS' COMMENTS

Reviewer #1 (Remarks to the Author):

The authors have responded satisfactorily to my comments and requests for revision.

Reviewer #3 (Remarks to the Author):

I thank the authors for satisfactorily responding to all the reviewer comments and recommend the manuscript for publication.